# N^6^-Methyladenosine RNA-Binding Protein YTHDF1 in Gastrointestinal Cancers: Function, Molecular Mechanism and Clinical Implication

**DOI:** 10.3390/cancers14143489

**Published:** 2022-07-18

**Authors:** Danyu Chen, Henley Cheung, Harry Cheuk-Hay Lau, Jun Yu, Chi Chun Wong

**Affiliations:** 1State Key Laboratory of Digestive Disease, Department of Medicine and Therapeutics, Institute of Digestive Disease, Li Ka Shing Institute of Health Sciences, The Chinese University of Hong Kong, Sha Tin, Hong Kong, China; chendanyu@link.cuhk.edu.hk (D.C.); henleycheung@link.cuhk.edu.hk (H.C.); harrylau@link.cuhk.edu.hk (H.C.-H.L.); 2CUHK Shenzhen Research Institute, The Chinese University of Hong Kong, Shenzhen 518172, China

**Keywords:** YTHDF1, N^6^-methyladenosine, gastrointestinal cancer, gastric cancer, colorectal cancer, liver cancer, oncogenic signaling, drug resistance, immunotherapy

## Abstract

**Simple Summary:**

N^6^-methyladenosine (m^6^A) is the most abundant internal modification in eukaryotic mRNA and plays a crucial role in the occurrence and development of diseases. YTHDF1 is the most powerful and abundant m^6^A-encoded RNA reader. In this review, we summarize the evidence of the involvement of YTHDF1 in gastrointestinal cancers, its molecular mechanisms of action, and therapeutic implications.

**Abstract:**

N^6^-methyladenosine (m^6^A) is the most abundant internal modification in eukaryotic cell mRNA, and this modification plays a key role in regulating mRNA translation, splicing, and stability. Emerging evidence implicates aberrant m^6^A as a crucial player in the occurrence and development of diseases, especially GI cancers. Among m^6^A regulators, YTHDF1 is the most abundant m^6^A reader that functionally connects m^6^A-modified mRNA to its eventual fate, mostly notably protein translation. Here, we summarized the function, molecular mechanisms, and clinical implications of YTHDF1 in GI cancers. YTHDF1 is largely upregulated in multiple GI cancer and its high expression predicts poor patient survival. In vitro and in vivo experimental evidence largely supports the role of YTDHF1 in promoting cancer initiation, progression, and metastasis, which suggests the oncogenic function of YTHDF1 in GI cancers. Besides, YTHDF1 overexpression is associated with changes in the tumor microenvironment that are favorable to tumorigenesis. Mechanistically, YTHDF1 regulates the expression of target genes by promoting translation, thereby participating in cancer-related signaling pathways. Targeting YTHDF1 holds therapeutic potential, as the overexpression of YTHDF1 is associated with tumor resistance to chemotherapy and immunotherapy. In summary, YTHDF1-mediated regulation of m^6^A modified mRNA is an actionable target and a prognostic factor for GI cancers.

## 1. Introduction

Gastrointestinal (GI) cancers refer to malignancies of the GI tract and auxiliary digestive organs. GI cancers remain the leading cause of cancer-related morbidity and mortality worldwide [1]. The most common GI cancers include gastric cancer (GC), colorectal cancer (CRC), and liver cancer. In addition, other digestive system cancers include esophageal cancer, small bowel cancer, pancreatic cancer, etc. Esophageal, gastric, and liver cancers are more prevalent in Asia than in the rest of the world, while Europe and North America have the highest burden of colorectal and pancreatic cancers [2]. Most patients with GI cancers present with locally invasive or distantly metastatic advanced tumors that preclude radical resection, and current treatments could not effectively control advanced disease. Novel therapeutic targets must thus be sought for GI cancers. Genetic and epigenetic alterations leading to the development of GI cancers have been well characterized through large scale sequencing projects. On the contrary, the role of the epitranscriptome, involving the chemical modification of RNA, in the development of cancer is just beginning to be explored. 

N^6^-methyladenosine (m^6^A) modification is a dynamic and reversible process and is the most pervasive modification in human mRNA. M^6^A methylation plays an important role in a wide variety of biological and pathological processes including cancer development. m^6^A regulators can be divided into three types: writers, erasers, and readers. The differential m^6^A profile is an emerging hallmark of cancer. Aberrant m^6^A deposition plays a key role in tumorigenesis in GI cancers by broadly altering gene expression. METTL3, a m^6^A methyltransferase, is up-regulated in gastric, colorectal, and liver cancers [3]. Besides, m^6^A modifications frequently involve key components of oncogenic signaling pathways, including WNT/β-catenin, phosphatidylinositol-3-kinase (PI3K)/Akt and mammalian target of rapamycin (mTOR) signaling pathways [4], thereby contributing to their aberrant activation in cancer. Understanding the function and molecular mechanism of the m^6^A-mediated epitranscriptome may thus unravel novel therapeutic targets and biomarkers for GI cancers. 

## 2. YTHDF1 Is a Major m^6^A Reader Involved in mRNA Translation

m^6^A readers are involved in many RNA processes, such as mRNA splicing, nuclear export, translation and decay in post-transcriptional regulation. YTH m^6^A RNA binding protein (YTHDF)-1 is one of the major m^6^A readers, of which it interacts with initiation factors to promote translation initiation in the cytoplasm [5]. Studies have shown that YTHDF1 plays an important role in the process of post-transcriptional modification by regulating the expression of genes involved in cancer, cell proliferation, cell migration and invasion, inflammation, immunity, and autophagy. 

m^6^A-modified mRNA interacts with YTHDF1 by placing its modified residues into the hydrophobic pocket of the YTH domain. YTHDF1 can recognize G(m^6^A) C and A(m^6^A)C RNA as ligands without sequence selectivity, thereby mediating the protein expression of m^6^A-modified target genes in health and disease conditions [6]. For example, YTHDF1 can increase the translation of m^6^A-modified mRNA, and this interaction is particularly pronounced in response to stress [5,7]. YTHDF1-mediated translation requires eukaryotic translation initiation factors (eIFs), including eIF3, eIF4E, and possibly eIF4G-dependent loop formation. Previous studies reported that YTHDF1 silencing leads to significantly downregulated eIF3A and eIF3B expression, and the translational efficiency of YTHDF1-targeted m^6^A transcripts decreased [8], indicating that YTHDF1 regulates translation efficiency in a m^6^A-dependent manner. YTHDF1 also enhances translation of the transcriptional regulator YAP by recruiting eIF3B to the translation initiation complex [9]. Of note, the knockdown of YTHDF1 has no effect on the m^6^A/A ratio of total mRNA, thus implicating that YTHDF1 regulates the association of its target mRNA with ribosomes rather than altering m^6^A modification level of mRNA. The role of YTHDF1 in mRNA stability cannot be completely ruled out, as an approximately 24% increase in the m^6^A/A ratio of mRNA was observed in YTHDF1 overexpressing cells [5]. As mRNA translation and degradation are closely correlated, YTHDF1 might preserve mRNA translation and slow down the rate of decay as a secondary effect.

Growing evidence has demonstrated the critical roles of YTHDF1 and its molecular mechanism in different cancers. For example, YTHDF1 interacts with the elongation factor eEF-2 in tumor cells, leading to m^6^A-induced translational elongation of Snail mRNA, a key transcription factor inducing epithelial-mesenchymal transition (EMT) [10]. Given that the importance of YTHDF1 in tumorigenesis has been increasingly recognized, here we review evidence supporting the roles and molecular mechanisms of YTHDF1 in different GI cancers. We also discuss the therapeutic potential of targeting YTHDF1 to improve treatment efficacy against GI cancer.

## 3. YTHDF1 in Gastrointestinal Cancers

YTHDF1 contributes to the tumorigenic behavior of cancer cells and facilitates a favorable tumor microenvironment. In this section, the functions and molecular mechanisms of YTHDF1 in GC, CRC, and hepatocellular carcinoma (HCC), the three most common GI cancers, are discussed (Figure 1 and Table 1).

### 3.1. YTHDF1 in Gastric Cancer

YTHDF1 is the most frequently mutated (>6%) m^6^A regulator in GC patients [11] and its overexpression is highly prevalent (>90%) among GC patients. From the GC cohort in The Cancer Genome Atlas (TCGA), YTHDF1 expression is significantly increased in early GC, and progressively rises further in advanced GC [12]. Moreover, high YTHDF1 expression is associated with more aggressive tumor progression and poor prognosis [11,13]. Notably, YTHDF1 overexpression is also subtype-dependent, as it was found to be more abundantly expressed in intestinal-type GC as compared to diffuse-type GC [14]. Collectively, these observations implicate the potential of YTHDF1 as an oncogenic factor in GC.

Multiple studies have demonstrated the oncogenic role of YTHDF1 in GC, which could be driven by overexpression or genetic mutation. Pi et al. [11] demonstrated that YTHDF1 knockdown attenuates the proliferation of GC cells in vitro and gastric tumorigenesis in vivo. Mechanistically, wildtype YTHDF1 promotes protein translation of a key WNT receptor, frizzled 7 (FZD7), in a m^6^A-dependent manner. Mutant YTHDF1 also induces FZD7 expression, causing the excessive activation of oncogenic WNT/β-catenin signaling pathway, thereby accelerating GC development [11]. Another target of YTDFH1 in GC is ubiquitin specific peptidase (USP)-14 [13]. USP14 could accelerate cell proliferation and migration in GC and induce resistance to cisplatin by promoting AKT/ERK signaling pathway [15]. Whereas the pharmacological inhibition of USP14 by IU1 abrogates the pro-tumorigenic effect of YTHDF1 in GC cells, thus indicating that the correlation between YTHDF1 and USP14 in contributing to gastric tumorigenesis.

YTHDF1 has been shown to promote GC through interplay with other m^6^A regulators. For example, YTHDF1 could function co-operatively with the m^6^A writer, methyltransferase 3 (METTL3), whereby METTL3 catalyzes m^6^A modification of sphingosine kinase 2 (SPHK2) mRNA, which in turn being recognized by YTHDF1. YTHDF1 stimulates SPHK2 translation in an eIF3A-dependent manner to induce gastric tumorigenesis [16]. Recent evidence indicates that YTHDF1 expression in cancer cells could modify the GC immune microenvironment. Bai et al., demonstrated that the loss of YTHDF1 in GC tumors provokes the complete disease remission in immunocompetent mice, but not in immunodeficient mice [17]. In immunocompetent mice, the loss of YTHDF1 stimulates the recruitment of mature dendritic cells (DCs) to GC with higher expression of major histocompatibility complex II and interleukin-12 secretion, thereby promoting CD4^+^ and CD8^+^ T cells infiltration. Mechanistically, RNA-sequencing revealed that the of loss YTHDF1 mediates the overexpression of interferon-γ receptor 1 and JAK/STAT1 signaling pathway in tumor cells, which may contribute to restoring sensitivity to antitumor immunity. Besides experimental evidence, YTHDF1 is inversely correlated with the expression of immune checkpoints including programmed cell death protein-1 (PD-1), PD-1-ligand-1 (PD-L1), and cytotoxic T-lymphocyte–associated antigen 4 (CTLA-4) in TCGA GC dataset, implying that YTHDF1 might negatively impact the efficacy of immune checkpoint blockade (ICB) [18]. Hence, these results suggest that YTHDF1 may be a potential target in gastric tumorigenesis and early diagnosis of GC. 

Epstein-Barr virus (EBV) is an oncovirus and its infection is associated with a subset of GC patients [19,20,21]. Recent evidence reported that YTHDF1 could suppress EBV replication by the recruitment of RNA degradation complexes including ZAP, DDX17, and DCP2 to promote RNA decay of EBV-related genes and thus down-regulating their translation [22]. While YTHDF1 primarily promotes the translation efficiency of m^6^A-modified mRNAs, the authors showed that YTHDF1 degrades m^6^A-modified viral transcripts in EBV-infected cells by the recruitment of ZAP, DDX17, and DCP2, which induces RNA uncapping during EBV infection of host cells. These findings thus offer novel insights into how YTDHF1 may impact the occurrence of EBV-associated GC through antagonizing EBV infection. Taken together, these results suggest that role of YTHDF1 might be subtype dependent in GC.

### 3.2. YTHDF1 in Colorectal Cancer

CRC is the most common GI cancer worldwide [23]. YTHDF1 is closely correlated to CRC of which the gain in DNA copy number of YTHDF1 is a frequent event in CRC patients (>60%), leading to YTHDF1 overexpression [24]. Among m^6^A regulators, YTHDF1 has the highest diagnostic value in distinguishing CRC from normal colon tissues in the TCGA cohort with AUC of 0.974 [25]. YTHDF1 is found to be upregulated even at precancerous adenoma stage [26]. Similar to GC, YTHDF1 expression rises progressively from the early to advanced CRC [25,26], and such gradual increase is positively correlated with clinical stage, lymph node metastasis and distant metastasis [27]. Liu et al., found that YTHDF1 is an independent risk factor for poor survival in CRC patients [28], concordant with its role as a pro-tumorigenic factor. 

Several studies have explored the functions and molecular mechanisms of YTHDF1 in CRC. Wang et al. [29] demonstrated that YTHDF1 promotes cell growth in CRC cell lines and primary organoids derived from CRC patients and is capable of promoting tumor metastasis in vivo. On the contrary, knockout of YTHDF1 markedly blunted colorectal tumorigenesis in carcinogen-induced CRC mouse model. Integrative m^6^A-seq, RIP-seq, ribo-seq and proteomics analysis led to the identification of RhoA activator ARHGEF2 as a key downstream target of YTHDF1 [29]. YTHDF1 binds to m^6^A-modified ARHGEF2 mRNA to promote its translation, and overexpression of ARHGEF2 could rescue the phenotypic effects of YTHDF1 depletion, verifying the role of YTHDF1-ARHGEF2 axis in CRC. Having demonstrated that YTHDF1-m^6^A-ARHGEF2 axis as a novel oncogenic signaling cascade in CRC, the authors constructed lipid nanoparticles (LNP) encapsulated with siRNA targeting ARHGEF2. LNP-siARHGEF2 was found to be effective in suppressing YTHDF1-induced oncogenic functions and liver metastases in CRC cells and animal models. 

The upregulation of WNT/β-catenin pathway, a key oncogenic pathway in CRC, is also closely correlated with YTHDF1 [24,29,30,31,32] (Figure 2). One study revealed that YTHDF1 exerts pro-tumorigenic effect by recognizing and promoting translation of m^6^A-modified FZD9 and WNT6 mRNA, leading to the aberrant activation of WNT/β-catenin signaling and ultimately promoting tumorigenicity and stem cell-like activity in CRC [24]. Similarly, Han et al. [31] showed that TCF4, DVL3, and FZD7, the three main components in WNT signaling, are direct targets of YTHDF1. YTHDF1-mediated the translation of these genes to activate WNT/β-catenin pathway and promote colonic cell stemness [31]. Consistently, colon-specific or colon stem cell-specific knockout of YTHDF1 impaired colorectal tumorigenesis and improved the survival in transgenic Apc^Min/+^ mice, therefore highlighting the importance of WNT as a target of YTHDF1 in CRC. Moreover, Jiang et el. identified that YTHDF1 boosts intestinal stemness by mediating translation of transcriptional enhancer factor TEAD1 [33]. Another study demonstrated the therapeutic potential of TEAD inhibitors against colorectal cancer stem cells by targeting YTHDF1-TEAD1 axis [34]. Additionally, YTHDF1 was found to rewire tumor metabolism by promoting the protein translation of glutaminase 1 (GLS1) by targeting the 3′UTR of GLS1 mRNA [35], which contributes to chemoresistance to cisplatin. Collectively, multiple downstream signaling molecules are involved in YTHDF1-mediated colorectal tumorigenesis. 

YTHDF1 also exerts a pro-tumorigenic effect by altering the tumor microenvironment. Inflammation is an etiological factor in the initiation and progression of CRC. Zong et al., demonstrated that YTHDF1-mediated translation of TNF receptor-associated factor 6 (TRAF6) is required for the activation of NF-κB and secretion of pro-inflammatory cytokines tumor necrosis factor (TNF)-α and IL-6 in intestinal cells [36]. Inflammasome NLRP3 is another target of YTHDF1, which provokes inflammatory injury stimulated by bacterial lipopolysaccharides [37], suggesting that YTHDF1 has pro-inflammatory effects in the colonic epithelium. Evaluation of the TCGA colon adenocarcinoma (COAD) dataset revealed that YTDHF1 high expression is associated with dampened adaptive immune response, cell killing, cytokine production and T cell activation [38]. Consistent with this, YTDHF1 high expression is negatively correlated with antitumor immune cell subtypes, including CD8^+^ T cells, M1 subtype macrophages, CD4^+^ T helper cells, DCs, natural killer cells, and natural killer T cells [38]. Altogether, these findings imply that YTHDF1 confers an immunosuppressive tumor microenvironment in CRC patients. The targeting of YTHDF1 to reactivate antitumor immunity may thus yield therapeutic benefits in combination with immunotherapeutics.

### 3.3. YTHDF1 in Hepatocellular Carcinoma

m^6^A has been implicated in various physiological and pathological processes in the liver [39,40,41,42,43]. HCC is the most common type of liver cancer, and YTHDF1 expression was found to be largely upregulated in HCC in multiple patient cohort studies with positive correlation to pathological stage [44,45]. Hence, high YTHDF1 expression is a major risk factor for predicting poor prognosis of HCC patients, including worse overall survival and progression-free survival [46,47]. Besides, high YTHDF1 expression is correlated with shorter recurrence-free survival after HCC resection [48,49]. YTHDF1 in conjunction with other m^6^A regulators, such as METTL3 and YTHDF2, could also be used to stratify high-risk HCC patients [50,51,52,53]. These studies underscore the potential role of YTHDF1 in metastatic progression of HCC.

Functional investigations have demonstrated that YTHDF1 exerts oncogenic functions by promoting proliferation of HCC cells in vitro and metastasis in mouse models [54]. Integrated analysis of RIP-seq/PARCLIP-seq and Ribo-seq in HCC cells identified that YTHDF1 depletion reduces translation efficiency of 413 genes involved in oncogenic pathways such as WNT and Hippo signaling. In particular, YTHDF1 accelerates the translation of FZD5 mRNA in a m^6^A-dependent manner, thereby promoting the WNT/β-catenin pathway. Multiple lines of evidence have implicated the role of YTHDF1 in EMT, a critical step in tumor metastasis. Two independent studies have shown that YTHDF1 promotes cell migration and invasion of HCC cells by inducing EMT and activating AKT signaling pathway [55,56]. In conjunction with METTL3-mediated m^6^A modification, YTHDF1 can directly induce the translation of Snail, a key transcription factor of EMT, thereby promoting liver cancer metastases [10]. Moreover, YTHDF1 has been shown to promote HCC metastasis in the context of inadequate radiofrequency ablation [57]. Sublethal heat stress was found to increase m^6^A modification and elevate YTHDF1 protein expression in HCC. Concomitantly, m^6^A-seq unraveled that m^6^A-mediated modification of EGFR mRNA is induced by sublethal heat stress, which in turn promotes its binding with YTHDF1, leading to the upregulation of EGFR translation to promote tumor metastasis. The combination of YTHDF1 silencing and EGFR inhibition dramatically inhibited HCC tumor metastasis after inadequate radiofrequency ablation in vivo, suggesting YTHDF1 as a potential therapeutic target for metastatic HCC. 

YTHDF1 can shape HCC tumor microenvironment [47,58] and potentiate the adaptation of HCC cells to promote their survival [59]. For example, Li et al., revealed that YTHDF1 plays a critical role in mediating protective autophagy in HCC cells, thereby allowing tumor cells to survive under the hypoxic tumor microenvironment [59]. Utilizing m^6^A-seq, proteomics and polysome profiling, the authors showed that YTHDF1 promotes the translation of m^6^A-modified autophagy-related genes (ATG)-2A and ATG14, thus facilitating the induction of autophagy in HCC cells. Apart from tumor cells, hepatic stellate cells (HSC) are an important component in HCC microenvironment as they can promote fibrosis. Interestingly, YTHDF1 activates autophagy in HSC by stabilizing m^6^A-modified BECN1 mRNA (responsible for regulating autophagosome formation), leading to HSC ferroptosis and the alleviation of fibrosis [60]. The differential roles of YTHDF1 between HCC tumor cells and HSCs may thus confer disparate outcomes for HCC development.

Recent studies have reported the association of YTHDF1 with antitumor immunity, of which high YTHDF1 expression in HCC contributes to immune evasion. For instance, high YTHDF1 expression is correlated with reduced immune cell infiltration in HCC [61]. Antitumor immune cells, including CD4^+^ T cells, γδ-T cells, and B cells, are all depleted in HCC with high YTDHF1 expression [62]. Tissue microarray validated that YTHDF1 overexpression in HCC is correlated with poor CD3^+^ and CD8^+^ T cell infiltration [61]. Xu et al., further showed that YTHDF1 is positively correlated with PD-L1 expression in HCC [47], implying that YTHDF1 may be a prognostic factor of HCC patients with poor response to ICB. Taken together, YTHDF1 regulates multifaceted processes in HCC and has great potential to be a therapeutic target and prognostic biomarker for HCC patients.

### 3.4. YTHDF1 in Other Gastrointestinal Cancers

Pancreatic ductal adenocarcinoma (PDAC) is one of the most aggressive malignancies among GI cancers. Multiple studies have shown aberrant upregulation of m^6^A RNA modification in PDAC tissues as compared to adjacent normal tissues [63,64]. In particular, YTHDF1 gene mutations are common in PDAC with frequent occurrence of in-frame deletions [65,66]. Huang et al., demonstrated that high YTHDF1 expression implicates a favorable prognosis in PDAC [67]. These observations infer that YTHDF1 might function as a tumor suppressor in PDAC. On the other hand, a few studies have also evaluated the prognostic significance of YTHDF1 in esophageal cancer, of which the high expression of YTHDF1 is correlated with shorter overall and progression-free survival [68]. Nonetheless, further research is needed to uncover and confirm the role and mechanistic functions of YTHDF1 in PDAC and esophageal cancer.

## 4. Clinical Implications of YTHDF1

GI cancer is among the most common malignancy worldwide with poor prognosis [69,70,71,72]. Therefore, there is an urgent need to improve diagnostic and prognostic tools, as well as to devise novel therapeutic strategy to improve the treatment of GI cancer. Increasing evidence indicates that YTHDF1 could dynamically regulate gene-specific translation in a m^6^A-dependent manner, leading to widespread deregulation of oncogenic signaling [31,73,74]. Here, we explore the current evidence showing that targeting YTHDF1 is a promising therapeutic approach to improve the efficacy of chemotherapy and immunotherapy against GI cancer.

### 4.1. The Impact of YTHDF1 on Chemotherapy

Chemoresistance, either primary or acquired, is the main cause for treatment failure and poor prognosis for cancer patients. Numerous mechanisms are involved in resistance to chemotherapy, including changes in drug disposition, adaptive responses, deregulation of cell death mechanisms, tumor microenvironment and epigenetic alterations. Recent work revealed that m^6^A-mediated epitranscriptome regulation contributes to cancer chemoresistance. For instance, m^6^A demethylases, such as FTO and ALKBH5, was shown to be essential for glioblastoma cancer stem cells and they mediate resistance to conventional chemotherapy and cancer recurrence in, respectively [75,76]. METTL3, a m^6^A methyltransferase, has been correlated with drug resistance of CD133^+^ GC stem cells [77]. Given that YTHDF1 is critical for determining the fate of aberrant m^6^A-modified mRNA, it is thus not surprising that YTHDF1 is closely associated with drug responsiveness. Nishizawa et al., showed that c-Myc induces YTHDF1 expression in CRC cells, and knockdown of YTHDF1 could sensitize CRC cells to chemotherapeutic drugs fluorouracil and oxaliplatin [27]. YTHDF1-mediated glutamine metabolism via GLS1 has been shown to promote cisplatin chemoresistance in CRC cells, while the combination of YTHDF1 silencing and cisplatin leads to synergistic effect in suppressing tumor growth [35]. YTHDF1 is also associated with the induction of cancer stemness in conjunction with m^6^A modification to contribute chemoresistance. The m^6^A-YTHDF1 axis has been implicated in the increased translation of tripartite motif-containing protein (TRIM)-29 to enhance the cancer stemness characteristics in cisplatin-resistant tumor cells [74]. Hence, targeting YTHDF1 may combat chemoresistance and alleviate the potential side effects of high-dose cisplatin [35].

Paradoxically, YTHDF1 was found to promote drug responsiveness towards a targeted therapeutic drug sorafenib, in HCC. Lin et al., showed that METTL3 promotes the transcription factor FOXO3 mRNA m^6^A methylation at the 3′UTR region and enhances its mRNA stability in a YTHDF1-dependent manner. FOXO3 negatively regulates the expression of autophagy-related genes (ATG3, ATG5, ATG7, ATG12, ATG16L1), thus inhibiting autophagy signaling and promoting sensitivity of HCC cells to sorafenib. Hence, the downregulation of the METTL3-YTHDF1-FOXO3 axis crucially contributes to sorafenib resistance, and targeting of YTHDF1 represents a novel therapeutic approach to enhance sorafenib response in HCC [41]. Given the highly diverse roles of YTHDF1 in promoting mRNA translation, the therapeutic effects of targeting YTHDF1 may vary in different cancers. In-depth research is therefore necessary to completely harness the m^6^A epitranscriptome for improving responses to cancer treatment.

### 4.2. The Impact of YTHDF1 on Immunotherapy

Emerging work has shown that immune checkpoint blockade (ICB) therapy is effective against advanced GI cancer, however; only a small subset of GI cancer patients could benefit from anti-PD-1/PD-L1 immunotherapy [78,79,80]. For example, in CRC, only patients with mismatch repair deficiency or microsatellite instability, comprising of ~5% of metastatic CRC, could be benefited from ICB therapy. Similarly low response rates were reported in GC or HCC patients receiving ICB therapy. Hence, there is an urgent need to identify factors that can modulate ICB responses. Various m^6^A regulators, such as METTL3 and ALKBH5, have emerged as key modifiers of the antitumor response by suppressing the function of antitumor CD8^+^ T-cells. On the other hand, YTDHF1 appears to be closely correlated with DCs in the tumor micro-environment. Genetic ablation of YTHDF1 in mice leads to reduced tumor growth associated with increased tumor infiltration by cytotoxic T cells, whilst simultaneously reducing infiltration of myeloid-derived suppressor cells (MDSC) [81]. Mechanistically, the deletion of YTHDF1 could promote cross-presentation of tumor-associated antigens by DCs, which activate CD8^+^ T cell-mediated adaptive immune response. As a consequence, ICB therapeutic response is significantly enhanced in YTHDF1-knockout mice as compared to wildtype mice. Consistently, our recent study implied the role of YTHDF1 in repressing DCs function in GC [17]. Loss of YTHDF1 in tumor cells leads to the recruitment of mature DCs in the tumor, which in turn promotes the infiltration of T helper cells and cytotoxic T cells, as well as the increased production of cytotoxic cytokines. These findings collectively imply that the targeting of YTHDF1 in GI cancer could reactivate antitumor immunity and potentiate the therapeutic effect of ICB therapy. Consistently, the low expression of YTHDF1 may serve as a marker of robust response towards ICB therapy in GI cancers. In summary, YTHDF1 might be an actionable target for improving the efficacy of ICB therapy in GI cancers. 

## 5. Conclusions and Perspectives

In summary, most of the studies reported thus far indicate that YTHDF1 is highly expressed in GI cancers and correlates with poor prognosis, implying that YTHDF1 functions as an oncogenic factor. Indeed, multiple studies have shown that YTHDF1 promotes the progression of GI cancers by the activation of WNT/β-catenin and PI3K/AKT/mTOR signaling pathways [11,24,55,82]. Mechanistically, YTHDF1 targets m^6^A-modified mRNA of key components of these pathways to promote their translation. Although many regulatory mechanisms have been discovered, there are still many unclear and controversial mechanisms due to the complexity of YTHDF1 regulation. Given the dynamic nature of m^6^A modifications, it is likely that the overall effect of YTHDF1 can vary in a context- and cell type-specific manner [17,83]. The exact role of YTHDF1 in GI cancers would be better elucidated using conditional, transgenic mice models that manipulate YTHDF1 in specific cell types, such as tumor cells or immune cells, in order to reveal the exact role of YTHDF1 in GI cancer development. Taken together, more research is needed to understand the role of YTHDF1-mediated epitranscriptome in the promotion of GI cancers. 

In terms of clinical application, YTHDF1 is reported to be an independent marker for the diagnosis and prognosis of GI cancer [17,29]. Interestingly, several studies have suggested that YTHDF1 and its downstream signaling pathways might serve as novel therapeutic targets against GI cancers, given the accumulating evidence showing its role in mediating drug resistance. Genetic depletion of YTHDF1 sensitizes GI tumors to chemotherapy and immunotherapy, suggesting that YTHDF1 antagonists can be potential adjuvants in GI cancer therapy. In particular, the targeting of YTHDF1 has been shown mediate a switch from immunological “cold” tumors to “hot” tumors, and thus presents great potential in combination with ICB therapy. Nevertheless, there is no pharmacological agent that could directly target YTHDF1. Hence, strategies must be sought to target YTHDF1 directly or to suppress its downstream targets. Given its close association with other m^6^A regulators, modulation of the m^6^A landscape might represent an alternative approach to targeting the oncogenic activities of YTHDF1. Altogether, the targeting of YTHDF1 represents a promising approach for the future management of GI cancers. 

**Table 1 cancers-14-03489-t001:** Characteristics of YTHDF1 in Gastrointestinal Cancers.

Cancer	Expression	Function	Molecular Targets	Reference
Esophageal Cancer	Increased	Oncogenic	NA	[68,84,85]
Gastric Cancer	Increased	Oncogenic	PD-1, PD-L1, CTLA-4	[13,18]
			FZD7/β-catenin	[11]
			SPHK2	[16]
			JAK/STAT1	[17]
Colorectal Cancer	Increased	Oncogenic	FZD9/WNT6/β-catenin	[24]
			c-MYC	[27]
			ARHGEF2	[29]
			TCF4/DVL3/FZD7/β-catenin	[31,32]
			TEAD1	[33]
			TRAF6, DDX60, NLRP3	[36,37,38]
Hepatocellular Carcinoma	Increased	Oncogenic	SNAI1	[10]
			PI3K/AKT/mTOR	[55]
			EGFR	[57]
			AKT/GSK-3β/β-catenin	[56]
			FZD5/β-catenin	[54]
			PD-L1, mTOR, WNT	[47]
			HIF-1α, ATG2A, ATG14	[59]
			TGF-β, WNT	[61]
Pancreatic Cancer	Decreased	Tumor Suppressive	NA	[63,64,65,66]

## Figures and Tables

**Figure 1 cancers-14-03489-f001:**
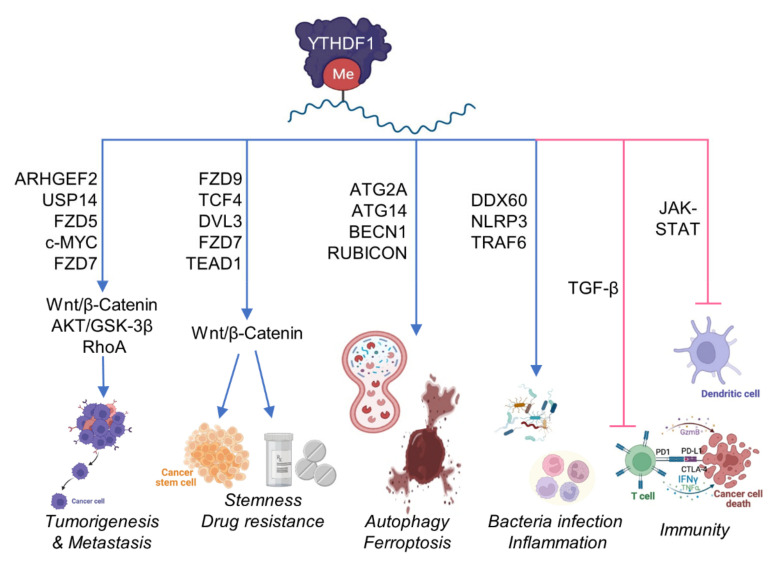
Multifaceted role of YTHDF1 in gastrointestinal cancers. YHTDF1-driven translation of m^6^A-modified gene targets has been shown to confer malignant phenotypes that are associated with cancer cell intrinsic properties (proliferation, metastasis, stemness and cell death) as well as its interaction with the tumor microenvironment (inflammation and immunity). Part of the figure is created with BioRender.com.

**Figure 2 cancers-14-03489-f002:**
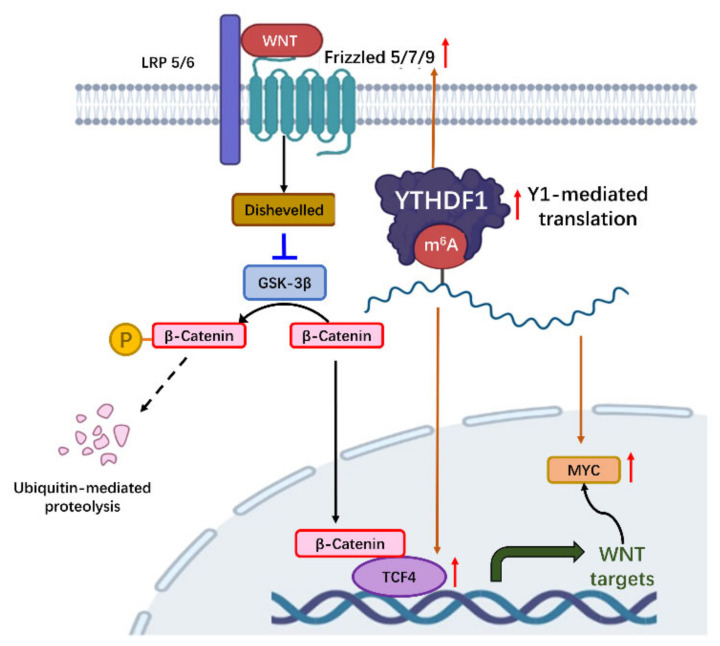
The function of YTHDF1 in promoting WNT/β-catenin signaling in gastrointestinal cancers. YHTDF1-driven translation of m^6^A-modified gene targets includes Frizzled 5/7/9, TCF4, and c-Myc. Increased Frizzled genes promoted the stability of β-catenin by inhibition of GSK3β, and elevated TCF4 enhanced β-catenin-mediated transcription. c-Myc, a key oncogenic target of WNT/β-catenin signaling, is also induced by YTHDF1. Part of the figure is created with BioRender.com.

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
