# Peer review of "N6-Methyladenosine RNA-Binding Protein YTHDF1 in Gastrointestinal Cancers: Function, Molecular Mechanism and Clinical Implication"

_cancers, 2022, doi:10.3390/cancers14143489_

Round 1

Reviewer 1 Report

m6A is the most prevalent internal RNA modification in mammals that regulates homeostasis and function of modified RNA transcripts. YTHDF1is a key regulator of m6A methylation in gastric cancer tumorigenesis. The topic of this review article is interesting. The references cited in this manuscript are updated. It would be helpful to add a section describing the role of m6A in gastric cancer tumorigenesis. This could further emphasize the importance of YTHDF1 in gastric cancer.

Author Response

Thank you for your positive comments. We have now added a section of the role of m6A in GI cancer in the introduction section, on line 53-61. 

Reviewer 2 Report

Chen et al. conducted a review study on YTHDF1, N6-methyladenosine RNA-binding protein. The comprehensive study not only discussed the molecular mechanism of YTHDF1 in cancer, but also its clinical significance in the treatment of cancer. The gastrointestinal (GI) cancers were the primary focus of this work. The work compiled the different molecular mechanisms of YTHDF1 from the previous studies and concluded the involvement of YTHDF1 in the various types of GI cancers.

Major comments:

1.     The introduction part mainly discusses the role of YTHDF1 in translation. However, in the last paragraph (line 72) authors mentioned that the role of YTHDF1 in translational relevance is poorly understood. Does it not sound contradictory?  

2.     Since this review is focused on the significance of YTHDF1 in GI cancers, I would recommend authors to write a paragraph about GI cancers in the introduction part.

3.     I see only a table and a figure in this review article. Authors might include at least one or two more figures describing the different cancer-related pathways YTHDF1 is involved.

Minor comments:

1.     The sentence in line 77 and line 85 is repeated.

2.     Reference missing in line 155

Author Response

Response to reviewer's comments:

Point 1: The introduction part mainly discusses the role of YTHDF1 in translation. However, in the last paragraph (line 72) authors mentioned that the role of YTHDF1 in translational relevance is poorly understood. Does it not sound contradictory?  

Response: We have revised this sentence accordingly on line 95-98.

Point 2: Since this review is focused on the significance of YTHDF1 in GI cancers, I would recommend authors to write a paragraph about GI cancers in the introduction part.

Response: We have added an overview on gastrointestinal cancers in the introduction section, on line 35-48.

Point 3: I see only a table and a figure in this review article. Authors might include at least one or two more figures describing the different cancer-related pathways YTHDF1 is involved. 

Response: We have now added an additional figures describing the molecular mechanism of YTHDF1 in WNT signaling pathway (Figure 2).